# An electrical analogy to Mie scattering

José M. Caridad[1,†], Stephen Connaughton[1], Christian Ott[2], Heiko B. Weber[2] & Vojislav Krstić[1,2]

Mie scattering is an optical phenomenon that appears when electromagnetic waves, in particular light, are elastically scattered at a spherical or cylindrical object. A transfer of this phenomenon onto electron states in ballistic graphene has been proposed theoretically, assuming a well-defined incident wave scattered by a perfectly cylindrical nanometer scaled potential, but experimental fingerprints are lacking. We present an experimental demonstration of an electrical analogue to Mie scattering by using graphene as a conductor, and circular potentials arranged in a square two-dimensional array. The tabletop experiment is carried out under seemingly unfavourable conditions of diffusive transport at room-temperature. Nonetheless, when a canted arrangement of the array with respect to the incident current is chosen, cascaded Mie scattering results robustly in a transverse voltage. Its response on electrostatic gating and variation of potentials convincingly underscores Mie scattering as underlying mechanism. The findings presented here encourage the design of functional electronic metamaterials.

[1] School of Physics, Centre for Research on Adaptive Nanostructures and Nanodevices (CRANN), AMBER at CRANN, Trinity College Dublin, College Green, Dublin 2, Ireland. [2] Department of Physics, Chair for Applied Physics, Friedrich-Alexander-University Erlangen-Nürnberg (FAU), Staudtstr. 7, Erlangen 91058, Germany. [†] Present address: Nanotech-Department of Micro and Nanotechnology, Technical University of Denmark, 2800 Kongens Lyngby, Denmark. Correspondence and requests for materials should be addressed to V.K. (email: vojislav.krstic@fau.de).

When transferring wave optical to electronic phenomena, graphene provides unique opportunities because its linear dispersion relation coincides qualitatively with that of photons. However, while in optical experiments typically a defined incident beam is chosen, in an electrical experiment inside a conducting material, electron waves are incident from all directions. One option to establish a conceptual link could be to define a beam in ballistic conditions, which has been successfully used in high-quality two dimensional semiconductors[1–3]. It is less obvious, how to make electrical analogues to optical experiments in a diffusive conductor.

We focus on Mie scattering[4], which is the scattering of a plane incident electromagnetic wave by a spherical or cylindrical object, rigorously described by the Maxwell equations. When considering an incident beam, the characteristic outgoing scattering profile puts emphasis on forward scattering, which is, however, widened[4].

Recent theoretical works have addressed the electronic analogy of Mie scattering in graphene at cylindrical potentials[5]. In essence, the Mie scattering is based on multiple Klein tunnelling at the potential wall, referred to as caustic motion[5,6]. Within the theory, considering a well-defined incident wave front, the spatial far-field scattering profile has been calculated, which can be effectively identified as electronic analogy to Mie scattering. Also, a device with a square array of quantum-dots in a ballistic graphene nanoribbon is calculated[7], which displays local deformations of the potentials, but the resulting conductance modulations loose significance with decreasing confinement and increasing separation of these quantum-dots as in our present 'optical' case (details below).

Although the existence of the Mie-like scattering was has been theoretically claimed, to-date no experimental demonstration was reported despite its fundamental interest as well as representing a root to realize purely electrical (two-dimensional) metamaterials which is relevant form the application viewpoint. Here we present experimental evidence for Mie scattering at cylindrical potentials in graphene demonstrating that this unique phenomena traditionally associated with light now appears in electronic transport. Since Mie-like scattering in electrical transport experiments undertaken on a single circular scatterer remains obscured, we introduced a trick, a two-dimensional canted array of such scatterers, that results in cascaded Mie scattering. The fingerprint of the cascaded Mie-like scattering is the generation of a transverse voltage, that is, results from the guiding of charge carriers to one of the edges of the sample, which we found to persist even at room temperature and under unfavourable conditions.

## Results

**Choice for canted arrays—cascaded Mie scattering.** We chose a similar arrangement of a square array of potentials as described in previous theoretical work[7], but with the all-important difference that the array is canted by an angle with respect to the macroscopic current direction (Fig. 1a,b). The difference becomes apparent when sketching Mie scattering profiles of electron waves of two cylindrical discs in series, based on a relativistic quantum-mechanical calculation (Supplementary Methods 1, Supplementary Figs 1–5) similar to refs 5,8–10. In an aligned geometry, the second disc is in the shadow of the first, and the profile is mirror symmetric (Fig. 1c). In the canted arrangement, however, this mirror symmetry is broken, as a disc is placed out of the shadow region such that it is exposed to strong intensity created by the first disc (Fig. 1d). In an array, this repeated mechanism is suited to provide electron trajectories that undergo cascaded Mie scattering when a finite current is biased.

These trajectories guide electrons to one side and the resulting drift electric field creates a robust transverse potential difference that can easily be measured (as illustrated in Fig. 1e).

Even under seemingly unfavourable conditions like imperfect scattering walls, diffusive motion in between the Mie scatterers and room temperature, this effect can be measured. Apparently, as justified below, cascaded Mie scattering supports ballistic remainders even under otherwise diffusive conditions. Note that in the strictly diffusive and non-relativistic regime, the square array geometry ensures an isotropic resistivity tensor (Supplementary Note 1, Supplementary Fig. 6).

**Transverse voltage as fingerprint of cascaded Mie scattering.** Our samples are mechanically cleaved graphene[11] (cf. Methods section) with an imposed square two-dimensional (2D) soft-potential superlattice (2D-SL Fig. 1e) with 350 nm periodicity ($d_{dot-dot}$), sufficiently large that it does not affect graphene's bandstructure[12] on $SiO_2$(300 nm)/Si substrates. Each soft-potential is realized by locally deposited metal on top of the closed graphene sheet, more precisely either a Ti or Pd circular dot of 100 nm diameter ($D$), a choice that shifts the local potential in the graphene sheet negatively or positively, respectively[13–15]. The size of each individual potential is much larger than the Fermi wavelength of graphene charge carriers, ensuring the operation of the device in a pure optical regime[5,6]. The level of disorder in graphene in between the dots was tuned by different types of substrate–surface treatment[16,17] and is directly correlated with the position of the sample's charge-neutrality point $V_{CNP}$[18–20]. To demonstrate cascaded Mie scattering, two types of devices were produced with the 2D-SL oriented at angles $\alpha_{2D-SL} = 30°$ and $0°$ relative to the biased current $I_b$ in x-direction (in Fig. 1a,b, cf. Methods section). For the full variety of samples, we first give an overview in Fig. 1f on a standard figure of merit (FoM) commonly used to analyse transversal signals[21] that is essentially the normalized transverse voltage at a relevant charge-carrier density/energy (the more accurate definition of this FoM is given below). The data reveal a strikingly clear distinction: for an aligned 2D-SL ($\alpha_{2D-SL} = 0°$), basically no transversal signal can be detected, whereas for a canted 2D-SL, a finite signal occurs. We propose that this transversal voltage $V_{trans}$ is the smoking gun indicating cascaded Mie scattering, a claim that is supported by several critical test that we carried out in detail. They include two crucial questions: (i) are the underlying symmetries reflected in the data, and (ii) can the functional dependence of the transverse voltage on experimental parameters be understood in the framework of the assumed mechanism?

The design of the experiment gives several handles to investigate the effect under symmetry inversion. A first and obvious choice is the inversion of the current direction ($I_b \rightarrow -I_b$) which can be compared with a directionally inverted illumination in an optical experiment. The expectations related to cascaded Mie scattering (Fig. 1c,d) suggest that intensity deviated in the $y$ direction is then guided to the $-y$ direction. In our electrical experiment, this corresponds to a charge accumulation on the opposite side. Indeed, the measured transversal voltage changes sign under this operation, as shown for Ti 2D-SL samples in Fig. 2a.

A further symmetry operation is an inversion of the electrostatics. The Klein tunnelling mechanism which underlies the Mie scattering is symmetric with respect to a simultaneous inversion of charge and electrostatic potentials. In our experiment, the charge can be inverted by a bottom gate. Independently, the potential landscape can be qualitatively inverted by a replacement of Ti, which decreases the local potential in the graphene sheet underneath, by Pd (resulting in an increase of the local potential). The approximate symmetry of

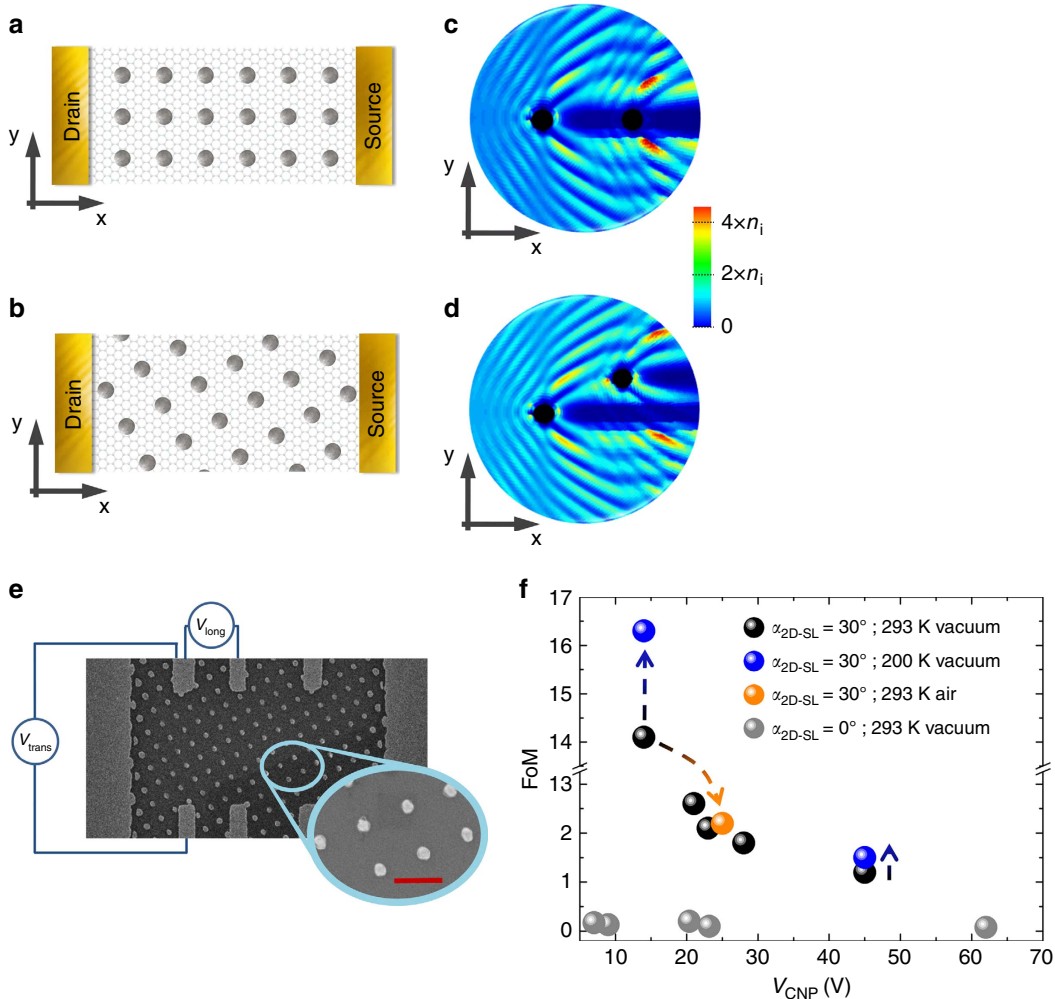

**Figure 1 | Cascaded Mie scattering in square potential 2D-SL.** Not-to-scale schematic of a top-view of graphene with (**a**) an aligned ($\alpha_{2D-SL} = 0°$) and (**b**) a canted 2D-SL ($\alpha_{2D-SL} = 30°$) on top contacted by source- and drain-electrodes (current flowing along $x$ axis). The grey discs indicate the potential-defining[13–15] metal dots on top of the graphene. In real samples the 2D-SL periodicity exceeds the graphene unit-cell size by orders of magnitude. Relativistic quantum-mechanical calculation of electronic density distribution for cascaded Mie scattering on two (**c**) aligned and (**d**) canted potentials (black, 100 nm diameter). In both cases the electronic wave-function is incident from the left along the $x$ axis (Fermi-energy 20 meV). For the canted arrangement an asymmetric re-distribution is found consisting of a pile-up of electronic (that is, charge carrier) density on one side (here top). This implies the generation of a transverse voltage along the $y$ axis. The scale bar is normalized to the incoming electronic density $n_i$ of $1 \times 10^{11}$ cm$^{-2}$. (**e**) A scanning electron image of graphene sample with a Ti 2D-SL rotated by $\alpha_{2D-SL} = 30°$ relative the direction of the traversing current ($I_b$). The 2D-SL has a 350 nm periodicity with dot diameter of 100 nm. Schematically indicated measurement of transverse and longitudinal voltage, $V_{trans}$ and $V_{long}$, respectively. Zoom: Rotated 2D-SL. Scale bar, 350 nm. (**f**) Standard FoM[21] for transverse signals analysis in electronic devices for aligned (grey symbols, $\alpha_{2D-SL} = 0°$) and canted ($\alpha_{2D-SL} = 30°$) 2D-SL samples studied with different charge-neutrality points ($V_{CNP}$). Strikingly, for all aligned 2D-SLs basically no transversal signal is observed, whereas for all canted arrangements a finite signal is detected. The FoM decreases with increasing $V_{CNP}$. Dashed arrows indicate change in FoM for samples on decreasing temperature change (black-blue) or exposure to ambient environment (black-orange).

simultaneous charge and potential inversion becomes apparent when plotting the transversal voltage $V_{trans}$ against the gate voltage $\Delta V_{gate}$ relative to the charge-neutrality point (Fig. 2b). Samples with Ti and Pd indeed respond oppositely on positive and negative gate voltages, which confirms the expectations. The charge-neutrality points of each sample have been independently derived from the gate-voltage ($V_{gate}$) dependence of the longitudinal resistivity $\rho_{long}$ (see as example Fig. 2c; Supplementary Note 2, Supplementary Fig. 7).

We now discuss the functional dependence of the data displayed in Fig. 2. The gate dependence of $V_{trans}$ is clearly non-monotonous, showing characteristic minima and maxima (*cf.* Fig. 2a). We interpret these data in the framework of a model that involves scattering at cylindrical potentials in graphene[22,23]. Such a sharp barrier invokes Klein tunnelling[11,24–26], and behaves similarly to

light scattering at a medium with negative refractive index[6]. The result for the cylindrical geometry is caustic propagation which leads to a Mie scattering profile. For the influence of the electrostatic potentials on this process it is sufficient to recall the energy dependence of Klein tunnelling. The Mie-scattered intensity of an individual dot as well as the cascaded Mie scattered intensity of the entire array is strictly proportional to the reflectivity at the potential step[23] and consequently the transverse voltage should follow this energy dependence.

In particular the reflectivity of electronic states is maximal when their energy matches with the upper edge of the potential step (see ref. 23 and Supplementary Method 1). It is however minimum at the bottom of the step. Therefore, the potential step height is a built-in energy-scale that should be found in the experimental gate-dependence of the transverse voltage.

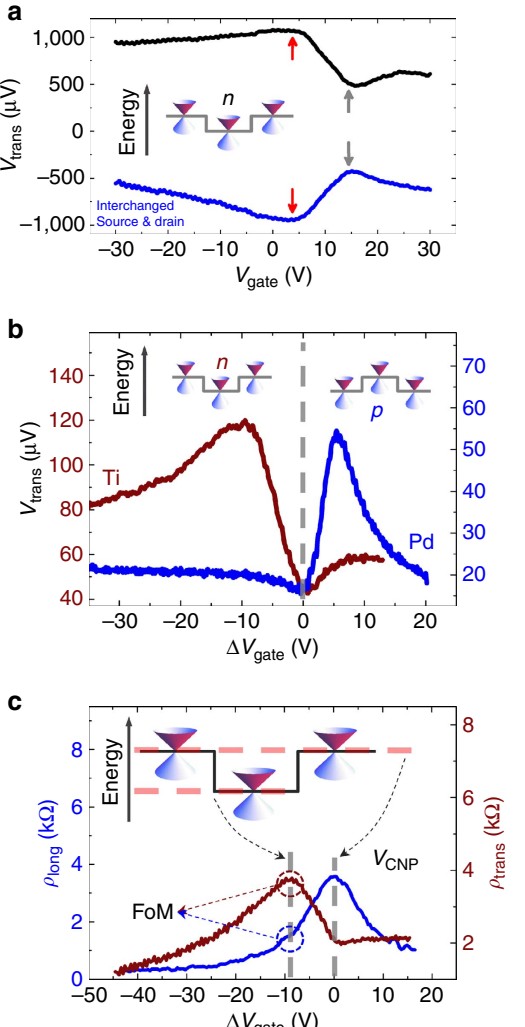

**Figure 2 | Symmetry of the transverse-signal generated by cascaded Mie scattering at room temperature.** (**a**) Transverse voltage $V_{trans}$ as function of gate-voltage $V_{gate}$ for a canted $\alpha_{2D-SL} = 30°$ 2D-SL sample (Ti dots). $V_{trans}$ shows two extreme points, a maximum (red arrow) at about 8 V and a minimum (black arrow) about 14 V. As expected, on reversal of the current direction $I_b \rightarrow -I_b$ (inverting source- and drain-electrodes), $V_{trans}$ reverses, too. (**b**) $V_{trans}$ for a Ti- (dark red) and Pd- (dark blue) defined canted 2D-SL sample. The transverse signals are plotted against $\Delta V_{gate}$, the gate-voltage relative to the charge-neutrality point. As expected from the energy dependence of Klein tunnelling which underlies Mie scattering, the minima of both signals are at the charge-neutrality point ($\Delta V_{gate} = 0$) and the maxima are found on opposite sides of $\Delta V_{gate}$. The opposite positioning of the maxima is a direct consequence of the inversion of the shift in local potential[13–15] (*cf.* insets) by choosing either Ti (*n*-type) or Pd (*p*-type) as dot material. The maximum is reached when the gate-voltage (carrier-energy) is at the maximum of the local potential shift. (**c**) longitudinal and transverse resistivity, $\rho_{long}$ (blue) and $\rho_{trans}$ (dark red), respectively, against $\Delta V_{gate}$ for a canted ($\alpha_{2D-SL} = 30°$) Ti-dots 2D-SL sample. $\rho_{long}$ shows the usual ambipolar dependence with a maximum at the charge-neutrality point $V_{CNP}$ as was observed in all samples with, both, canted and aligned geometry. That is, our 2D-SLs do not affect graphene's bandstructure[12]. $P_{trans}$ has the identical characteristics as its corresponding transverse voltage, as observed in all samples investigated. Dashed circles indicate the relevant values for determining the FoM.

A first remarkable experimental feature (Fig. 2b) is that the minimum of both curves coincides with the charge-neutrality points of each sample ($\Delta V_{gate} = 0$)[23,25]. This allows for a straight-

forward comparison of the maximum position in $|V_{trans}|$ with the theoretically calculated potential heights induced by Pd and Ti on top of graphene. For Pd the analysis of the data in Fig. 2b results in a potential height of $+0.06\,eV \pm 0.01\,eV$ which matches very well with the $+0.08$ to $+0.09\,eV$ reported[27,28] with no free parameter (*cf.* Methods section). For Ti, experimentally we determined a potential height of $-0.12\,eV \pm 0.01\,eV$ which provides still a reasonable match to the $-0.23$ to $-0.28\,eV$ stated in literature[27–29]. We conclude that the gate-dependence can qualitatively be related to the energy dependence of Klein tunnelling which provides another decisive and independent check for the claim that the transverse voltage indicates cascaded Mie scattering. Fig. 2 motivates our definition of the FoM aforementioned, it is the ratio between the maximum transverse resistivity $\rho_{trans} = V_{trans} \cdot I_b^{-1}$ and the corresponding $\rho_{long}$ at the same gate-voltage (charge-carrier density/energy) in accordance to standard analysis approaches of transversal signals[21] (*cf.* Fig. 2c).

**Theoretical analysis of the cascaded Mie scattering.** After all experimental tests were underscoring the cascaded Mie scattering scenario, we turn to the theoretical analysis.

Mie scattering at single (potential) discs[5,6,23] as well as at aligned arrays[7] has been calculated in detail. We pick up this concept and apply it to two neighboured discs which are misaligned to the incident current direction to elucidate the mechanism of cascaded Mie scattering. An example of the results was already presented in Fig. 1d.

Technically, the scattering matrix approach has been employed (Supplementary Method 1) to calculate the scattering of an incoming plane wave off a sequence of two discs (Supplementary Figs 2–4). Figure 3a displays the far-field scattered electronic intensity reflecting the distribution of the electronic density along the intersect in *y*-direction after scattering from the two discs at angles $\alpha_{2D-SL} = 30°$ and $0°$ for three different gate-voltages. Both, the oscillatory behaviour and the shadow close to $y = 0$ are characteristic features of Mie scattering[4,5]. As expected, for the aligned geometry ($\alpha_{2D-SL} = 0°$) no transversal asymmetry arises in the electronic density. This is in contrast with canted geometry ($\alpha_{2D-SL} = 30°$ as in the experiment) in which a pronounced asymmetry is found. This asymmetry can be evaluated when summing up the far-field scattered intensity independently for positive and negative values of *y*. Their difference is a measure for the deviation of the incident beam due to the canted geometry.

Thus, these canted two discs are the simplest element that supports the generation of a transversal current component or, depending on the measurement circuit a transversal voltage. When further cascading more and more of these units, the incoming current experiences a transversal drift resulting in a charge-accumulation on one side (*cf.* Fig. 1d), measured as transversal voltage (as illustrated in Fig. 1e).

This scheme can be extended to a qualitative calculation of the energy-dependence of the transverse voltage. As shown in Fig. 3a, the magnitude of the intensity imbalance is minimal at a Fermi energy close to the charge neutrality point meanwhile it is maximal at the potential barrier, a behaviour fully coinciding with experimental observations (Fig. 2). It can be traced back to Mie scattering occurring in individual dots (Supplementary Method 1). To underpin this, Fig. 3b plots the total scattered far-field intensity ($I_{scatt}$) as a function of $\Delta V_{gate}$, for the potential heights $+0.06\,eV$ and $-0.12\,eV$ as found in our experiments for only one circular potential (further details in Supplementary Method 1, Supplementary Fig. 1). In addition, the (scaled) transverse voltage measured at a Ti and a Pd sample are plotted (no scaling with respect to the abscissa). The experimental step-feature is

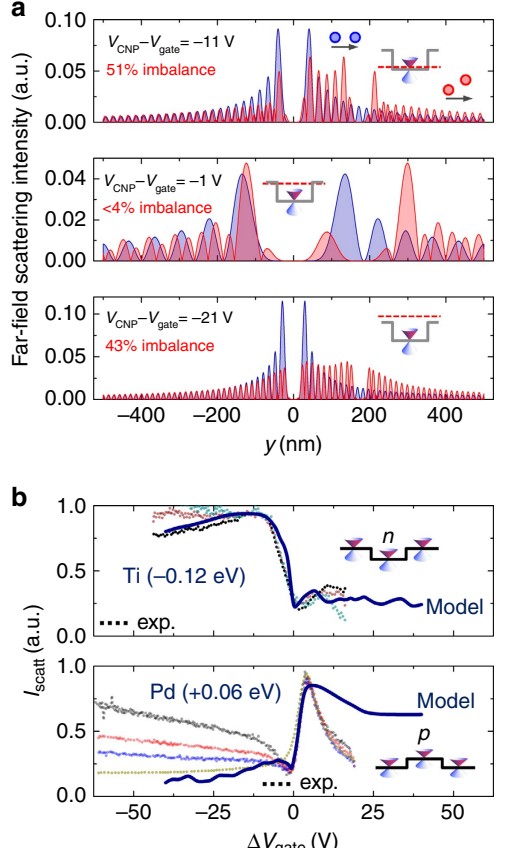

**Figure 3 | Modelling cascaded Mie scattering of electronic waves and resulting transverse-signals.** (**a**) From top to bottom, the cross-sections of the electronic far-field scattering intensity, 100 nm after scattering from two negative-valued circular potentials (100 nm diameter, *cf.* Fig. 1c,d), along the $y$ axis of a 2D-SL unit cell (centre $y = 0$) for gate-voltages $V_{gate} = -11$ V, $-1$ V and $-21$ V relative to the charge-neutrality point $V_{CNP}$. Canted potential arrangements ($\alpha_{2D-SL} = 30°$) are in red and aligned ($\alpha_{2D-SL} = 0°$) are in blue. For all canted cases an imbalance of intensity (integration over corresponding axis intercept) relative to the unit-cell centre is found depending on the position of the Fermi-level (gate-voltage). No imbalance is found for any aligned case. The establishment of an imbalance (difference in intensity) is equivalent to a transverse-voltage generation across the unit-cell. The oscillatory behaviour is characteristic for Mie scattering. Insets: Schematic of Fermi-energy position (dashed line) relative to the potential well (grey line); circles and arrows indicate the orientation of the two circular potentials relative to the incident electronic wave (current) direction for $\alpha_{2D-SL} = 0°$ and 30°. (**b**) Total scattered far-field intensity $I_{scatt}$ for a negative-valued (top) and a positive-valued (bottom) single circular potential as a function of $\Delta V_{gate}$ with potential magnitudes as extracted from our experiments using Ti and Pd dots. Dotted lines are scaled experimental transverse-voltage data for a Ti-defined and a Pd-defined 2D-SL sample taken at different biases (range 0.5–6 mV, Supplementary Fig. 4, Supplementary Method 2). Our calculations reproduce remarkably well for both types of potentials the minimum and maximum and their positions in all the displayed experimental transverse voltages. Systematic deviations towards higher gate-voltage magnitudes are attributed to higher-order scattering processes not considered in our calculations.

remarkably well reproduced by the calculations for both types of potential heights. For higher voltages there is a systematic deviation that we relegate to our future work.

The agreement between the experimental and theoretical results is very convincing, however, it is worth mentioning that

the experiments are undertaken under seemingly unfavourable conditions. The latter include imperfect circular shape of the lithographically defined dots, diffusive motion of the charge carriers between the dots as well as incoherent electronic motion at room temperature. Recent experiments have however shown that Klein tunnelling based devices can result in macroscopic observables under similar conditions[30].

**Persistence of cascaded Mie scattering**. For completion, we extend our study towards more favourable as well as unfavourable conditions. We present results on single samples under various conditions. The first case is realized by lowering the temperature from room temperature to 200 K under vacuum ($\sim 3 \times 10^{-6}$ mbar). As expected, the FoM (that is insensitive to gate-offset) increases as observed in two independent graphene sheets of very different $V_{CNP}$ (14 and 45 V) by the same amount of 15–20% (Fig. 1f, Supplementary Note 3). The second case has been tested by venting the vacuum chamber at room temperature such that the sample is exposed to ambient atmosphere. The FoM is significantly lowered by about 84%, which is also reflected in the increase of $V_{CNP}$ and corresponds to a reduction of the minimum (Coulomb) mean-free-path[18–20] from 140 down to 100 nm (Supplementary Table 1). This perfectly agrees with our general finding in Fig. 1f that the FoM (the development of the transverse voltage) reduces as the charge neutrality point is shifted to higher voltages (Supplementary Notes 4–6, Supplementary Figs 8–10). We would like to note that the propagation of the charge carriers within a dot-potential needs to be of wave-like nature to ensure a caustic motion which then results into the Mie-like scattering at an individual dot (*cf.* Fig. 1)[5,6]. Given the dot diameter in our experiments to be 100 nm, a minimum mean-free-path of about half the dot diameter would be therefore required to observe Mie-like scattering. This minimum scattering distance is well below our experimentally determined minimum mean-free-path range (77–140 nm, Supplementary Table 1) in agreement with the observation of a transverse voltage in all our devices with canted arrays of dots. Considering all the points stated above and the circumstance that our experimental mean-free-paths are in all devices below the closest dot to dot separation (250 nm), we conclude that the observed cascaded Mie scattering signal is not fully insensitive to unfavourable conditions but still robust enough to be detectable in everyday table-top experiments.

## Discussion

Altogether, we provide an experimental scheme that translates a well-known optical phenomenon, the Mie scattering, to its electrical analogue. A canted array of cylindrical potentials is sufficient to demonstrate the Mie-like scattering of the charge carriers and, importantly, to convert graphene into an electrical metamaterial.

The resulting electrical phenomenon of a transverse voltage using cascaded Mie scattering is rooted in the relativistic character of the charge carriers and is, therefore, unique to all (2D) systems with comparable electronic properties. Interestingly, the transverse voltage effectively results from the guiding of charge carriers to one of the edges of the sample. This reminds on the Hall effect were a magnetic field forces a steady-state transverse drift of moving charge carriers, however, in our system with the difference that no magnetic field is applied. From this viewpoint, our system represents an easy tuneable relativistic charge–carrier guiding mechanism based on scattering at room-temperature which is an appealing opportunity for the development of new device functionalities involving Dirac-fermion systems.

## Methods

**Device preparation.** Our samples are mechanically cleaved graphene[11] with an imposed square 2D soft-potential superlattice (2D-SL, Supplementary Figs 1–3) realized by lithographically defined circularly shaped metal dots on top (Fig. 1e). The performance of exfoliated graphene devices highly depends on the substrate preparation[16,17,31]. We deliberately tuned the $SiO_2/Si$ substrates to obtain varying device qualities allowing to investigate the persistence of the optics-like guiding at different charged-impurity levels: hydrophobically rendered $SiO_2/Si$ substrates without plasma treatment to minimize the presence of random charge impurities, and untreated $SiO_2/Si$ substrates. Ti (30 nm thick film) and Pd (25 nm thick film) are the contacting and 2D-SL-defining material deposited on top of the graphene by standard electron-beam material deposition under vacuum ($\leq 3 \times 10^{-6}$ mbar). Two types of devices were produced with the 2D-SL oriented at angles $\alpha_{2D-SL} = 30°$ and $0°$ relative to the direction of the current flowing through the sample (cf. Fig. 1a,b for illustration).

**Electrical measurements.** Unless otherwise specified, samples were measured in vacuum ($< 10^{-5}$ mbar) and at room temperature. A constant bias DC voltage was applied between the source and drain contacts and each, the longitudinal voltage and transverse voltage were measured independently in two separate measurements, but simultaneously with the corresponding current traversing the sample (cf. Fig. 1e for illustration). Through this any signal mixing is excluded. Moreover, we note that this electrical setup is commonly used in graphene subjected to local potentials[32]. In addition, the measurement of the FoM was carried out at a constant current of $I_b = 200$ nA. The charge-carrier energy with applied gate-voltage was obtained by first determining the carrier-density through the conversion factor[19] $0.072 \times 10^{12}$ V$^{-1}$ cm$^{-2}$ taking the $SiO_2$ thickness into account (300 nm) and successive insertion into graphene's dispersion relation considering the square-root-dependence of the charge-carrier wave-vector on the carrier-density[17]. With this, the potential height due to the local metal dots can be directly read-out as difference between maximum and minimum position in the transverse signal.

**Raman measurements.** Exfoliated graphene layers to be identified as monolayers through Raman microscopy (Supplementary Fig. 11). Raman spectra were measured at room temperature with a Renishaw micro-Raman spectrometer using a $\times 50$ objective (laser spot $\sim 4 \mu m^2$) and a 633 nm wavelength excitation laser. The spectral resolution of the apparatus was $\sim 2$ cm$^{-1}$. The calibration was carried out using the Rayleigh and Si (521 cm$^{-1}$) bands. Spectra acquisition time was of the order of few minutes keeping the power on the samples below 1 mW to avoid sample heating

**Data availability.** The authors declare that the data supporting the findings of this study are available within the article and its Supplementary Information Files.

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

## Acknowledgements

V.K. and H.B.W. acknowledge support through the SFB 953 'Synthetic Carbon Allotropes' funded by the Deutsche Forschungsgemeinschaft (DFG). J.M.C. acknowledges financial support from the EC Graphene FET Flagship. Also, V.K. acknowledges support by the Science Foundation Ireland (PI-award 08/IN.1/I1873, CSET 08/CE/I1432).

## Author contributions

V.K. designed the experiments and supervised the entire study. J.M.C. mechanically exfoliated graphite and characterized resulting graphene layers. J.M.C. and S.C. prepared the devices and carried out the electrical measurements. J.M.C. carried out the scattering matrix calculations. C.O. and H.B.W. developed and utilized the resistor network model. V.K., H.B.W. and J.M.C took lead in manuscript write-up.

## Additional information

**Competing financial interests:** The authors declare no competing financial interests.

