## [Peer Review File · Nature Communications]

Reviewers' comments:

Reviewer #1 (Remarks to the Author):

The paper by Carida et al addresses the scattering of electronic waves in cleaved graphene with superimposed soft-potential quantum dots arranged in a square superlattice. This is a very interesting and timely problem with potential technological relevance for future graphene-based nano electronic, plasmonic and photonic devices. The focus is on an electrical analogue of the optical Mie scattering phenomenon in graphene with cylindrical potentials; an effect that has been predicted and elaborated theoretically before. It's marvellous that this proposal could be realised in the present work, even under the unfavourable conditions of room-temperature ballistic transport. The nice idea was to use a canted quantum dot array with respect to the incident macroscopic current direction, leading to cascaded Mie scattering which gives reason to a transverse voltage. In this manner Mie scattering and Klein tunnelling at the array of potential walls could be convincingly demonstrated by a simple tabletop experiment. Being a theoretician, I cannot examine the specifications of the experiment in great detail, but I'm impressed by the results presented as well as by the data analysis and discussion. Thereby the experiment nicely shows the symmetries of the (Mie-scattering Klein-tunnelling) problem (current inversion, simultaneous inversion of charge and electrostatic potentials) and explains the functional dependence of the transverse voltage on the experimental parameters. The fact that the gate-dependence can be related to the energy-dependence of the Klein tunnelling also corroborates the quoted cascaded Mie-scattering scenario. On the whole, the agreement between the experimental findings and the theoretical predictions is impressive. Beyond that it appears that the observed Mie scattering signal is largely robust when the experimental conditions change for the worse (e.g., at room temperature and ambient atmosphere), which for sure is advantageous having applications in mind. Summing up, the paper provides strong evidence for its conclusions, the results are interesting, important and should influence the specific field. I recommend its publication in Nature Communications.

Reviewer #2 (Remarks to the Author):

The authors provide an experimental demonstration of the electrical analog of Mie scattering in modulated graphene. This experimental work is based on theoretical predictions of Ref.5. This paper contains several important findings and innovations. First, the authors find that Mie-like scattering by an array of biased graphene patches persists even in the diffusive transport regime. That is an important distinction from the theoretical calculation of Ref.[5], where electron scatterings were neglected. Second, the authors realized the importance of the tilted geometry, where the electron current flows at an angle to the principal axes of the 2D crystal. The tilted geometry makes the system anisotropic, thereby enabling a voltage drop in the y-direction when the current flows in the x-direction. The results are convincing, and I believe that the paper is suitable for Nature Communications after addressing the following issues.

(a) The authors mention that the main effect (the absence of the transverse voltage) vanishes under purely diffusive conditions. They should clarify the physics of this vanishing. Specifically, I can see two things happening under the conditions of electron scattering. First, the electron transport inside circular defects is no longer wave-like, and Mie resonances are expected to vanish. Second, the transport between the defects is affected. It would be nice to discuss which of the effects is more important and, therefore, how one would estimate the minimum scattering distance that still allows

for resistance anisotropy to develop.

(b) The authors should provide a scale for the color bar in Fig.1. Otherwise I cannot tell by how much the electron density is modulated.

Response to the Reviewers' comments:

Reviewer #1 (Remarks to the Author):

The paper by Carida et al addresses the scattering of electronic waves in cleaved graphene with superimposed soft-potential quantum dots arranged in a square superlattice. This is an very interesting and timely problem with potential technological relevance for future graphene-based nano electronic, plasmonic and photonic devices. The focus is on an electrical analogue of the optical Mie scattering phenomenon in graphene with cylindrical potentials; an effect that has been predicted and elaborated theoretically before. It's marvellous that this proposal could be realised in the present work, even under the unfavourable conditions of room-temperature ballistic transport. The nice idea was to use a canted quantum dot array with respect to the incident macroscopic current direction, leading to cascaded Mie scattering which gives reason to a transverse voltage. In this manner Mie scattering and Klein tunnelling at the array of potentials walls could be convincingly demonstrated by a simple tabletop experiment. Being a theoretician, I cannot examine the specifications of the experiment in great detail, but I'm impressed by the results presented as well as by the data analysis and discussion. Thereby the experiment nicely shows the symmetries of the (Mie-scattering Klein-tunnelling) problem (current inversion, simultaneous inversion of charge and electrostatic potentials) and explains the functional dependence of the transverse voltage on the experimental parameters. The fact that the gate-dependence can be related to the energy-dependence of the Klein tunnelling also corroborates the quoted cascaded Mie-scattering scenario. On the whole, the agreement between the experimental findings and the theoretical predictions is impressive. Beyond that it appears that the observed Mie scattering signal is largely robust when the experimental conditions change for the worse (e.g., at room temperature and ambient atmosphere), which for sure is advantageous having applications in mind. Summing up, the paper provides strong evidence for its conclusions, the results are interesting, important and should influence the specific field. I recommend its publication in Nature Communications.

Our response: We thank the reviewer for his/her very positive comments on our work and recommending it for publication in Nature Communications.

Reviewer #2 (Remarks to the Author):

The authors provide an experimental demonstration of the electrical analog of Mie scattering in modulated graphene. This experimental work is based on theoretical predictions of Ref.5. This paper contains several important findings and innovations. First, the authors find that Mie-like scattering by an array of biased graphene patches persists even in the diffusive transport regime. That is an important distinction from the theoretical calculation of Ref.[5], where electron scatterings were neglected. Second, the authors realized the importance of the tilted geometry, where the electron current flows at an angle to the principal axes of the 2D crystal. The tilted geometry makes the system anisotropic, thereby enabling a voltage drop in the y-direction when the current flows in the x-direction. The results are convincing, and I believe that the paper is suitable for Nature Communications after addressing the following issues.

Our response: We are thankful to the Reviewer for judging our work being suitable for Nature Communications. We have answered in detail (see below) the two comments of the Reviewer and have amended our manuscript and the supplementary material accordingly.

(a) The authors mention that the main effect (the absence of the transverse voltage) vanishes under purely diffusive conditions. They should clarify the physics of this vanishing. Specifically, I can see two things happening under the conditions of electron scattering. First, the electron transport inside circular defects is no longer wave-like, and Mie resonances are expected to vanish. Second, the transport between the defects is affected. It would be nice to discuss which of the effects is more important and, therefore, how one would estimate the minimum scattering distance that still allows for resistance anisotropy to develop.

Our response: We first address the last point of the author, providing the rationale for the minimum scattering distance that would still allow for the resistance anisotropy to develop. This will also shine light on the two previous points, no longer wave-like transport within circular defects (in main manuscript referred to as circular dots) & transport affected between circular defects. For the development of the resistance anisotropy the Mie-like scattering at an individual circular defect (*cf.* Refs. 5,6) is required leading to the needed redistribution of the electronic density (*cf.* Fig. 1c and d). In other words, the far-field current shows in this case a pronounced forward scattering at an individual circular defect.

To ensure this local Mie-like scattering, the scattering distance within a circular defect should be such that the charge-carriers can propagate following a caustic (*cf.* Refs. 5,6) which implies a wave-like transport. The result of this caustic propagation is the aforementioned electronic density redistribution/forward scattering, that is, the Mie-like scattering. Therefore, as conservative estimate, the scattering-distance should not be smaller than half the diameter of the circular defect. In our experiments the circular defects have a diameter of 100 nm. That is, the resistance anisotropy is expected in our type of devices to vanish for a minimum scattering distance ≤ 50 nm. This is consistent with our experimentally extracted minimum mean-free path ($l_{m,CNP}$) at the charge-neutrality-point V_{CNP} , Table S1 of our Supplementary Material, for all devices showing a resistance anisotropy.

We found a range of $77 \text{ nm} \leq l_{m,CNP} \leq 140 \text{ nm}$ for these devices consistent with our afore outlined reasoning that the minimum scattering distance should be larger than half of the diameter (50 nm) of a circular defect.

This estimation of the minimum scattering distance answers also the first point of the Reviewer on no longer having wave-like transport inside circular defects: If the minimum mean free path is (well) below half the diameter of a circular defect, then electron transport is expected not to be governed by wave-like propagation. This in turn implies that the Mie-like scattering at an individual circular defect vanishes and hence no canted arrangement of circular defects would lead to a resistance anisotropy.

On the 2nd point of the Reviewer on affected transport between defects with increased scattering as well as which of the effects is more important: As rationalised above, if no Mie-like scattering occurs locally at a circular defect then no resistance anisotropy can develop in devices with canted arrays of circular defects. Thus, Mie-like scattering at individual circular defects is the key prerequisite for the appearance of the resistance anisotropy. The role of diffuse motion in between the circular defects is certainly more subtle. Diffusion brings along a loss of momentum information, which (weakly) counteracts the directional Mie scattering. Yet, the re-equilibration of the (forward-scattered) charge density beams generated through Mie-scattering at a single circular defect (*cf.* Fig. 1c,d) will also be damped by diffusion in between the circular defects. That is, the re-equilibration of the charge density is obstructed to some extent therefore working towards the retention of the charge density beams which, when cascaded, lead to the resistance anisotropy. This is consistent with our experimental findings (*cf.* Fig. 1f) that the magnitude of the resistance anisotropy increases with decreasing values of the charge-neutrality-point V_{CNP} , that is, equivalently, increasing minimum mean free path. Another aspect of diffuse motion makes our experiments particularly elegant: Diffusive electronic propagation in between circular defects suppresses the possibility of interference of local Mie-scattering events which happen at different circular defects, such that the picture of cascaded Mie-scattering becomes adequate. Again, our experiments corroborate this overall picture as for all our devices showing a resistance anisotropy the minimum mean free path (77 nm to 140 nm) is below the shortest distance between two neighbouring circular defects (250 nm). Summarising, our experiments show that the scattering in between the circular defects play a comparably lesser role. The interplay of all the aforementioned effects is very subtle and deserves in principle in-depth investigation, however, which we relegated to further future work.

We have added and commented the main results of our discussion here in the revised manuscript text on page 9 in the 2nd paragraph, lines 12 to 22 from top. The added text reads “*We would like to note that the propagation of the charge carriers within a dot-potential needs to be of wave-like nature to ensure a caustic motion which then results into the Mie-like scattering at an individual dot (cf. Fig. 1).*^{5,6} *Given the dot diameter in our experiments to be 100 nm, a minimum mean-free-path of about half the dot diameter would be therefore required to observe Mie-like scattering. This minimum scattering distance is well below our experimentally determined minimum mean-free-path range (77 to 140 nm, supplementary Table S1) in agreement with the observation of a transverse voltage in all our devices with canted arrays of dots. Considering all the points stated above and the circumstance that our experimental mean-free-paths are in all devices below the closest dot to dot separation (250 nm), we conclude that the observed cascaded Mie scattering signal is not fully insensitive to unfavourable conditions but still robust enough to be detectable in everyday table-top experiments.*”

(b) The authors should provide a scale for the color bar in Fig.1. Otherwise I cannot tell by how much the electron density is modulated.

Our response: We have specified in line 11 of the figure captions of Fig. 1 the scale of the color bar, and have also added to the color bar the associated numbering. The additional sentence in the figure caption reads "*The scale-bar is normalised to the incoming electronic density of $1 \times 10^{11} \text{ cm}^{-2}$.*".

REVIEWERS' COMMENTS:

Reviewer #2 (Remarks to the Author):

I am happy with these changes, the manuscript can now be accepted for publication.

Point by point response to referees comments

Reviewer #2 (Remarks to the Author):

I am happy with these changes, the manuscript can now be accepted for publication.

Our response:

We thank the reviewer for his efforts and recommending our manuscript now for publication in Nature Communications.